# Interspecific ICSI for the Assessment of Sperm DNA Damage: Technology Report

**DOI:** 10.3390/ani11051250

**Published:** 2021-04-26

**Authors:** Jana Rychtarova, Alena Langerova, Helena Fulka, Pasqualino Loi, Michal Benc, Josef Fulka

**Affiliations:** 1Institute of Animal Science, 104 00 Prague, Czech Republic; helena.fulka@gmail.com (H.F.); benc.michal@gmail.com (M.B.); fulka.josef@vuzv.cz (J.F.J.); 2GENNET, 170 00 Prague, Czech Republic; alena.langerova@gennet.cz; 3Institute of Experimental Medicine, 142 20 Prague, Czech Republic; 4Faculty of Veterinary Medicine, University of Teramo, 64100 Teramo, Italy; valutatorearea07@unite.it; 5Faculty of Natural Sciences, Constantine the Philosopher University in Nitra, 94974 Nitra, Slovakia

**Keywords:** sperm, oocyte, DNA damage

## Abstract

**Simple Summary:**

The long-term storage of biological material (sperm, oocytes, embryos, etc.) is essential not only for animal breeding programs, human ART, and basic biology, but also for rescuing endangered animal species and other technologies. Whilst sperm storage is almost perfected for some species (bovine, human, mouse), many problems remain in others. In our contribution, we present a simple approach that can be used for the rapid evaluation of DNA damage level in sperm nuclei after freezing. This approach can be useful especially in those cases when the amount of frozen biological material is limited and permits much higher flexibility when modifying chosen preservation approaches.

**Abstract:**

Xenogenic mammalian sperm heads injected into mouse ovulated oocytes decondense and form pronuclei in which sperm DNA parameters can be evaluated. We suggest that this approach can be used for the assessment of sperm DNA damage level and the evaluation of how certain sperm treatments (freezing, lyophilization, etc.) influence the quality of spermatozoa.

## 1. Introduction

Mammalian spermatozoa only exceptionally fertilize intact or zona-free oocytes of another mammalian species. The only exception seems to be the combination of hamster zona-free oocytes, into which mammalian spermatozoa of other species penetrate quite frequently when in vitro fertilization (IVF) is used. After the penetration, the sperm heads decondense in a foreign cytoplasm and form pronuclei. Nevertheless, in this case the spermatozoa must be highly motile [1,2]. High motility, however, does not mean that the sperm DNA is not damaged. At the same time, it also does not mean that non-motile sperm cannot produce functional pronuclei. Indeed, it has been demonstrated that immotile and even dead spermatozoa, for example, when lyophilized or recovered from frozen cadavers, can produce normal offspring when injected into the same species oocyte [3,4]. On the other hand, the motile spermatozoa with damaged DNA (i.e., after chemotherapy) can fertilize the oocyte but subsequent development is compromised [5]. Typically, these spermatozoa decondense in the cytoplasm and form pronuclei with delayed DNA replication; further cleavage is abnormal, and micronuclei can be detected in the cytoplasm of the two-cell embryos [6,7].

Many factors can damage sperm DNA. This can occur already during spermiogenesis by excessive ROS (reactive oxygen species) generation, by anticancer drug treatments, smoking, air pollution, etc. [8]. Beside this, sperm DNA can be damaged even when conventionally used techniques are inappropriately applied, e.g., ICSI (intracytoplasmic sperm injection) [9] or during the sperm heads preparation for ICSI [10,11]. Specific attention must also be paid to some sperm storage procedures such as cryopreservation and lyophilization [12].

There are many approaches that can be used for the evaluation of sperm DNA damage. Typically, these approaches need a relatively high number of spermatozoa for evaluation; therefore, they cannot be used, for example, for oligospermic samples [13]. 

Assessing the intensity of γH2AX labeling is a commonly used approach for the evaluation of DNA damage in somatic cell nuclei. Histone H2AX is rapidly phosphorylated at sites of DNA double-strand breaks (DSBs), and this phosphorylated H2AX (γH2AX) then recruits numerous repair proteins [14,15,16].

Our “technology report” presents a relatively simple approach that can be used for sperm DNA damage evaluation, especially in those cases where sperm numbers are very low.

## 2. Materials and Methods 

Mouse oocytes were isolated from oviducts of superovulated B6D2F1 females (PMSG 5 IU i.p. with hCG 5 IU i.p. approximately 44 h post PMSG; Intervet, Boxmeer, The Netherlands). The ovulated oocytes were isolated from oviduct ampullae after about 14–15 h post hCG and incubated in M2 medium supplemented with hyaluronidase (0.1%). This incubation facilitates the removal of cumulus cells by vigorous pipetting. The oocytes were then cultured briefly in KSOM (Millipore, Prague, Czech Republic) at 37 °C in a humified incubator (5% CO_2_ in air) and then used for intracytoplasmic sperm injection (ICSI). 

For ICSI, we used fresh or cryopreserved goat, ram, bovine, pig, horse and rooster spermatozoa from Accredited Insemination Stations. The frozen samples were thawed in a water bath (37 °C), washed several times with M2 medium and then used for ICSI. As this is a methodological paper, for ICSI we used two extremes—fresh (highly vital—motile with sharp contours) or badly frozen, i.e., mostly immotile (~10% motile)/no sharp contours sperm samples. 

ICSI was performed essentially as described by Yoshida and Perry [17]. Briefly, the oocytes (10–20) and spermatozoa were placed into 10 µL of M2 covered with paraffin oil on the lid of a 10 cm Petri dish. Isolated sperm heads (whole rooster sperm) were injected into oocytes with a piezo injector PMAS-CT150 (Tsukuba, Ibaraki, Japan) on the inverted microscope Olympus stage IX 71 (Olympus, Prague, Czech Republic), magnification 40×.

Immediately after the sperm head injection, the oocytes (zygotes) were washed several times in KSOM and cultured in it for ~9 h as described above. Then, the interspecific zygotes were inspected under the inverted Olympus IX 71 microscope with Hoffman optics, selected, and suitable oocytes transferred into M2—their zonae pellucidae were removed by acid Tyrode solution and zona-free zygotes were fixed in 4% paraformaldehyde for 15 min at room temperature. The fixed samples were then kept in PBS (Phosphate Buffered Saline) in a refrigerator before labeling.

The fixed samples were first permeabilized in 0.2% Triton X-100 (Sigma, Prague, Czech Republic) in PBS for 10 min at room temperature (RT) and then blocked overnight in 1% BSA,(Bovine Serum Albumin) 0.1% TX-100 in PBS in a refrigerator. The samples were then incubated in the same solution with the first antibody for 1 h: γH2AX 1:200 (Abcam, Cambridge, UK), and then, after extensive washing, in PBS/BSA in the appropriate secondary antibody (Alexa Fluor donkey anti rabbit, 1:800, Jackson ImmunoResearch, Ely, UK) for 2 h. After extensive washing, the samples were mounted in Vectashield and evaluated under the fluorescence microscope Olympus BX 61 (Olympus, Prague, Czech Republic).

Each ICSI combination was repeated at least three times with more than 50 oocytes injected for every species. 

If not stated otherwise, all chemicals were purchased from Sigma, Prague, Czech Republic.

## 3. Results

The survival of injected oocytes was almost absolute and only exceptionally they died during or immediately after injection. More than 90% of oocytes were activated when injected with the sperm, i.e., they extruded the second polar body and contained at least the maternal (female) pronucleus (fPN). With the exception when rooster spermatozoa were used (see below), in about 75% of activated oocytes two PNs were visible already under the stereomicroscope (summarized in Table 1). No intact sperm heads were detected in activated oocytes with a single pronucleus, when stained with Hoechst. These oocytes were discarded and not used for further labeling.

First, we wanted to know the response of different species sperm heads to the mouse oocyte cytoplasmic environment. The responses are graphically depicted in Figure 1. When goat, ram, bovine and horse spermatozoa were injected into mouse oocytes, their sperm heads decondensed and formed paternal (male) pronuclei (mPNs) that were always larger than maternal (mouse) pronuclei (fPNs). The second polar body was extruded in all cases. fPNs were located in the vicinity of second polar bodies, whilst the mPNs were located more distantly. Paternal and maternal PNs contained distinct nucleolus precursor bodies (NPBs). Interestingly, pig sperm also activated the oocytes with the formation of normal fPNs, but the paternal PN was always slightly smaller. A more interesting situation was the combination: rooster x mouse. Here, the sperm did not form PNs and only slightly decondensed, whilst fPNs were always present. What is also interesting is that highly motile rooster spermatozoa swam very actively in the mouse cytoplasm for a few minutes post injection.

After labeling against γH2AX, only a minor signal in mPNs was observed when fresh sperm samples were used, and essentially no labeling in fPNs was observed—this was also observed for the pig. On the other hand, very intensive labeling in mPNs (sperm) was observed when improperly frozen samples were used for injection (Figure 2). No labeling was observed in almost unchanged rooster sperm heads and evidently these sperm heads did not respond to the mouse oocyte cytoplasm. Here, we labeled several oocytes with a lamin B antibody (not described in detail), and as evident from Figure 2, the nuclear lamina was present only in fPNs. However, we cannot rule out that the antibody does not recognize avian lamin B.

In conclusion, we demonstrate that interspecific ICSI can be used for the evaluation of sperm DNA damage. For practical purposes, it is, however, necessary to set the limits of damaged DNA which cannot be repaired by the oocyte.

## 4. Discussion

There are several approaches that can be used for the evaluation of sperm quality. Classically, sperm motility, viability, concentration and gross morphology serve as the basic criteria to evaluate the quality of semen. With the advent of more sophisticated approaches in ART (ICSI, ROSI, etc.) that can overcome immotility and low sperm numbers, more specific methods are being used (for example, the evaluation of acrosome morphology, sperm membrane intactness, etc.). Among them is the evaluation of DNA integrity, i.e., sperm DNA damage. Several approaches can be used for the assessment of DNA integrity—acridine orange test, sperm chromatin structure assay, chronomycin A3, aniline and toluide blue staining, in situ nick translation, terminal deoxy nucleotidyl transferase mediated dUTP nick end labeling assay (TUNEL), sperm chromatin dispersion, single cell gel electrophoresis or comet assay, etc. [19]. These approaches typically need a relatively high number of spermatozoa and show relatively low sensitivity. 

In general, heavy sperm DNA damage leads to embryo development arrest. When sperm DNA is less damaged, embryo development is impaired, embryos show genomic instability, and if offspring are born, they may exhibit certain abnormalities such as altered adiposity and regulation of glucose in females [20,21].

Logically, the relevant experiments studying the relationship between sperm DNA damage and further embryo development were conducted mostly in rodents, where a relative high quality of cells (sperm, oocytes) can be obtained. Besides this, in vivo embryo techniques are almost perfected here. This is more complicated in domestic animals, where, for example, the oocytes are in vitro produced, ICSI needs to be perfected and if transferable embryos are produced they cannot be transferred into uteri in similarly high numbers as in rodents [22]. Moreover, the evaluation of DNA damage in intact spermatozoa is not very accurate because their chromatin is tightly packed with protamines. In our “Technology Report”, we took the advantage that mammalian sperm heads decondense in foreign cytoplasm and form pronuclei, in which protamines are replaced with histones [17]. When bovine, ram, goat, horse and human sperm heads were injected into mouse ovulated oocytes, paternal pronuclei were always larger than maternal ones. Interestingly, in the pig the situation was quite the opposite, but even in this case, the level of sperm DNA damage could be more precisely evaluated than in intact spermatozoa. We observed a very interesting and unexpected situation when rooster spermatozoa were injected into mouse oocytes, when only a slight sperm head decondensation was observed. It has been supposed that a different P1/P2 ratio of different protamine compositions may be responsible for altered sperm head decondensation and subsequent formation of pronuclei [23]. These hypotheses are interesting and need to be confirmed by further studies.

Our approach can be used especially in those cases when the sperm number is quite low—oligospermic samples or in some specific cases, for example, when cadavers are recovered, as demonstrated already for somatic cell mammoth nuclei which were injected into mouse oocytes [24]. 

In general, in about two days, we can answer the question of whether the chosen sperm sample is suitable for further use or not or eventually if a given sperm storage method is convenient for sperm conservation or if it is necessary to modify it. Logically, it is necessary to find a threshold intensity of fluorescence, which will indicate that above that level the sperm DNA cannot be repaired by the oocyte, as it is well known that the oocyte contains DNA repair activities that are able to correct a low level (less than 8%) of damaged sperm DNA [25]. 

## 5. Conclusions

In conclusion, in our contribution, we present a new approach for the evaluation of sperm DNA damage after ICSI into ovulated mouse oocytes. We are well aware that many factors may influence the quality of sperm, including the season, the method of sperm collection, freezing/thawing/lyophilization procedures, etc. The interspecific ICSI gives us an almost immediate possibility for rapid evaluation of the sperm samples and subsequent modifications of the above-mentioned approaches.

## Figures and Tables

**Figure 1 animals-11-01250-f001:**
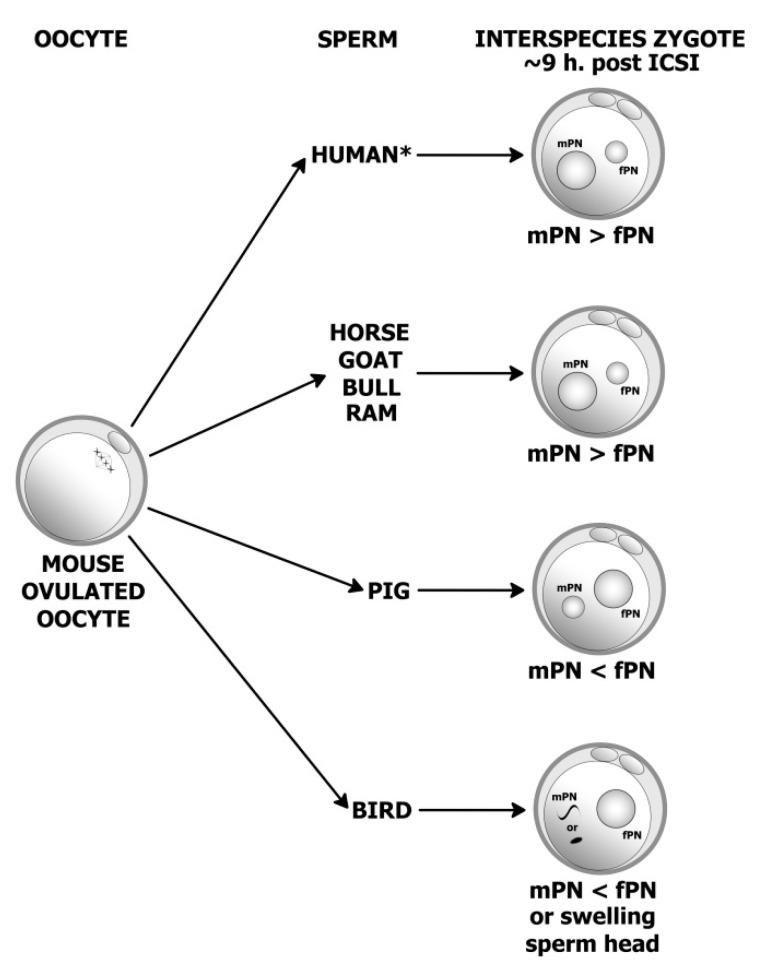
Decondensation of foreign sperm heads in ovulated mouse oocytes. Paternal pronuclei (mPNs) were always larger than maternal (female fPNs) when human, bull, horse, goat and ram spermatozoa were injected into ovulated mouse oocytes (* previous work: Fulka et al. [18]) and samples were evaluated approximately 9 h post ICSI. On the other hand, in the pig the mPNs were smaller than fPNs. When rooster spermatozoa were used for ICSI, no sperm head decondensation occurred (eventually only minor sperm head swelling was observed).

**Figure 2 animals-11-01250-f002:**
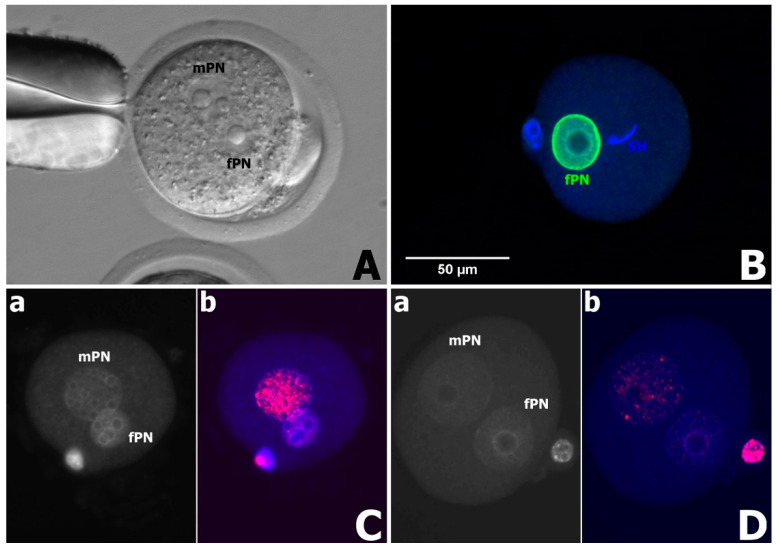
Representative pictures demonstrating the behavior of sperm heads after interspecific ICSI. (**A**) Goat sperm in mouse oocytes forms large mPN with several nucleoli. Maternal, female—fPN is always smaller. (**B**) Rooster sperm (SH) does not respond to mouse oocyte cytoplasm whilst the maternal (fPN) is normally formed. (**C**) Heavily damaged sperm DNA (mPN) exhibits very intensive labeling against ɣH2AX—(**a**) DNA staining with Hoechst (mPN—sperm, fPN—maternal pronucleus), (**b**) parallel picture showing very intensive ɣH2AX signal in mPN. (**D**) Only weak labeling against ɣH2AX is detected over mPNs when fresh spermatozoa are used for ICSI ((**a**)—Hoechst staining, (**b**)—ɣH2AX labeling).

**Table 1 animals-11-01250-t001:** Xenogenic sperm head injection into mouse ovulated oocytes.

Sperm Origin	No. of Oocytes Injected/Survived (%)	Activated with Both PNs and 2PB *	γH2AX in mPNs (Fresh Sperm) **	γH2AX in mPNs (Frozen Sperm) **
horse	52/49 (94%)	37 (76%)	15 (+/−)	20 (+++)
goat	67/62 (93%)	48 (77%)	20 (+/−)	24 (+++)
bull	51/49 (96%)	37 (76%)	17 (+/−)	18 (+++)
ram	63/59 (94%)	45 (76%)	20 (+/−)	22 (+++)
pig	60/58 (97%)	45 (78%)	22 (+/−)	21 (+++)
Rooster ^+^	50/49 (49%)	0	n.a.	n.a.

* The remaining oocytes extruded the second polar body and contained only a single PN. No intact sperm heads were detected in the oocyte cytoplasm after Hoechst staining. ** Some injected oocytes were lost during labeling. ^+^ Whole rooster sperm was injected into the mouse oocyte cytoplasm. +/− Only minor labeling was detected over the mPNs with essentially no labeling in fPNs. +++ Heavy labeling over mPNs and no labeling in fPNs. n.a. Not applicable.

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
