# Peer review of "Interspecific ICSI for the Assessment of Sperm DNA Damage: Technology Report"

_animals, 2021, doi:10.3390/ani11051250_

Round 1

Reviewer 1 Report

The work as a whole is confusing. The title refers to a technical report on the assessment of DNA integrity using ICSI among different species. The introduction does not include variation among the species tested. The methodology should be clearer in terms of the experimental design of the species used. The results describe differences in size between the pro-nuclei, which was not the objective of the study, but it is an interesting result that should be highlighted. The discussion section does not explain the results found in the manuscript. The discussion should explain the differences found between the species in terms of the size of the pro-nuclei or the reason why the rooster sperm was inferior to the other species in the formation of the male pro-nucleus. It is succinct and does not explain the experiments done at work. Finally, the conclusion points more to the ICSI technique than to the DNA integrity analysis technique. The manuscript needs to be revised as a whole for what it was intended to present. 

Author Response

Note for reviewers: Corrections in the text are marked in red

Response to reviewers

Rev. 1.

First, we would like to thank Rev. 1 for his/her comments. Shame that his/her notes are not very specific. So not easy to reply  -  work as a whole is confusing???

In fact, we are well aware that this is a preliminary work (proof of principle) which needs to be expanded and perfected. For these reasons we have used two extremes – fresh vs. heavily damaged spermatozoa. Logically, in this case hardly we can expect some variations among species.

It is evident that the evaluation of DNA damage of much more accurate than in intact spermatozoa. Beside this, this approach enables us to characterize (in the first mitosis) the sperm karyotype, chromosome breaks, micronuclei and so on. This everything we wish to study and to compare with some others approaches.

Certainly, studying the size of PNs is an important point. But rather academic question.

Reviewer 2 Report

I would suggest some changes mainly in results and discusión.

MATERIALS AND METHODS

Jackson ImmunoResearch instead of Jackson Immuno Research

RESULTS

There is no table where you can see the total number of ICSIs performed using fresh or cryopreserved sperm from goat, ram, bovine, pig, horse and rooster.

Inaccurate results are indicated, such as: “The survival of injected oocytes was almost absolute”, “in about 75 % of oocytes …”

In the text that explains Figure 1 it is indicated " samples were evaluated approximately 8 h post ICSI” and the image shows 9h post ICSI

In Fgure 2, picture C you should indicate what is “b

DISCUSSION

Too many "approaches", try to use synonyms and other expressions.

Try to compare the results with other papers. Really the first paragraph looks like a review.

Permafrost?

Any idea to find a limit (intensity of fluorescence) which will indicate that above that level the sperm DNA cannot be repaired by the oocyte?

Do you suggest the use of the technique for any specific species?

My advice would be to rethink Results and Discussion,

Author Response

Note for reviewers: Corrections in the text are marked in red

Rev. 2.

Many thanks for valuable comments. We did our best to modify the manuscript as suggested. Marked in RED.

M & M …. Jackson ImmunoResearch …. modified

Results …. Table is now included

Inaccurate results ….. modified

Text to Fig. 1 …. modified as suggested

DISCUSSION is now modified. Please, take this Technology Report as a preliminary work (proof of principle) which needs to be expanded and perfected. For this reason, it was difficult to compare our approach to some other evaluation methods - especially in domestic animals. This is discussed in the corrected manuscript. As mentioned in the text, we must find a limit in DNA damage. We know, this will be hard in domestic animals. We do believe that this approach can be used in those species, where the sperm head decondense in a foreign cytoplasm.

Reviewer 3 Report

This is an interesting study. Using mouse oocytes, in combination with interspecies ICSI, as a means to evaluate the DNA damage extent in the sperm samples is a novel idea. The authors conducted experiments using sperm samples from several species. In this regard, the work provides valuable information to the community. 

I have several comments:

(1) Please present the summary data of the work. For example, a Table illustrates species, number of oocytes injected, quantitative measure of H2X staining. 

(2)  The authors "assumed" that there are more DNA damages in the "badly frozen" sperm samples. This should be evidence based, using one or several methods authors described in the Discussion, given that sperm number is not a issue. Authors should provide data supporting this, and correlate such data with the ICSI work. Only in this way, the conclusion can be supported. At the minimum, this should be done thoroughly in one species. 

(3) Any thoughts on the the "exception" of the rooster sperm in the work? Please include discussion on this. 

Author Response

Note for reviewers: Corrections in the text are marked in red

Rev. 3

Many thanks for valuable comments

The results are now summarized in Table 1., we do believe that results are more clear now. We do not measure the intensity of ɣH2AX, because we have used two extremes. Fresh/Heavily damaged. It is a preliminary study which needs to be expanded. Logically, we will evaluate different approaches used for sperm storage in our further experiments. Beside this, we wish to study sperm karyotypes, micronuclei, chromosome breaks, etc.

3/ This is now discussed in the text. Probably different protamines. Interesting question – but more academic.

Round 2

Reviewer 1 Report

The manuscript was improved after include a table in results section and discussion complements.

My best regards.

Author Response

Many thanks for your efforts and time.

Reviewer 3 Report

(1) Materials and Methods, "The samples were thawed in water bath (37oC), several times washed with M2 medium and then used for ICSI. "

Please add "frozen", so it reads "The frozen samples were thawed..."

(2) Materials and Methods, "As this is a methodological paper, for ICSI we have used two extremes – fresh (highly vital – motile) or badly frozen, i.e. almost immotile sperm samples."

There is no data to support that the sperm samples are "almost immotile". Also, immotile does not have to be due to DNA damage. Can authors find support (that frozen sperm have much higher levels of DNA damage comparing to fresh ones) from some published literature?

(3) Please move the paragraph that introduces rH2AX in the Discussion part to the Introduction. 

(4) Discussion, "Logically, it is necessary to find a limit (intensity of fluorescence) which will"

Change to "limit" to "threshold".

Author Response

Many thanks for Rev. 3 comments. Alle of them improved the manuscript.

ad 1. red in the text

ad 2. changed, red in the text - mostly immotile now - only about 10% motile.

I have checked the literature again, I did not find in it that information - frozen sperm/higher level of DNA damage

ad 3. moved - red

ad 4/ changed - threshold now